# Beam Scanning and Capture of Micro Laser Communication Terminal Based on MEMS Micromirrors

**DOI:** 10.3390/mi14071317

**Published:** 2023-06-27

**Authors:** Xuan Wang, Junfeng Han, Chen Wang, Meilin Xie, Peng Liu, Yu Cao, Feng Jing, Fan Wang, Yunhao Su, Xiangsheng Meng

**Affiliations:** 1Key Laboratory of Space Precision Measurement Technology, Chinese Academy of Sciences, Xi’an 710119, China; 2Xi’an Institute of Optics and Precision Mechanics, Chinese Academy of Sciences, Xi’an 710119, China

**Keywords:** free-space optical communication, MEMS micromirror, laser scanning and capture

## Abstract

With the development of space laser communication and the planned deployment of satellite Internet constellations, there is a growing demand for microminiature laser communication terminals. To meet the requirements of size, weight and power (SWaP), miniaturized terminals require smaller drive components to complete on-orbit scanning and capture, which must be fast and efficient to enable satellite laser communication networks. These miniaturized laser communication terminals are highly susceptible to the impact of the initial pointing accuracy of the laser beam and microvibrations of the satellite platform. Therefore, this paper proposes a laser scanning-capture model based on a Micro-electromechanical Systems (MEMS) micromirror that can provide a fast, large-scale scanning analysis. A scanning overlap factor is introduced to improve the capture probability under the influence of microvibrations. Finally, experimental analysis was carried out to verify the effectiveness of the proposed model, which can establish a theoretical basis for future ultra-long-distance microspace laser communication.

## 1. Introduction

Compared to traditional RF technology, free-space optical communication offers significant advantages in data rate performance, cost-effectiveness, and increased security [1]. As such, it plays an important role in space-ground integrated networks. At the same time, CubeSat satellites are becoming increasingly popular in low Earth orbit networks due to their low cost, quick response times, potential for constellation formation, and ability to perform tasks that a single large satellite cannot [2,3]. Since the German Aerospace Center began to focus on the research and development of small satellite optical communication terminals more than 10 years ago, this field has attracted more and more attention from major research institutions: NASA, the Jet Propulsion Laboratory, European Space Agency, and National Institute of Information and Communication Technology of Japan [4,5].

In 2008, the German TerraSAR-X and the United States NFIRE satellites marked a new phase in space laser communication by successfully establishing the first coherent link over a distance of more than 5000 km at a rate of 5.6 Gbit/s [6,7]. The TerraSAR-X satellite is equipped with the LCT135 laser terminal developed by DLR and has become the standard configuration of the European Data Relay Satellite System (EDRS). LCT135 has made significant contributions to the establishment of high-orbit backbone satellite networks and low-orbit satellite constellations. After more than 30 years of research and technological accumulation, TESAT has developed a range of new laser communication terminal (LCT) products [8,9].

However, establishing optical communication links between CubeSats is a daunting task. The laser transmission link requires very strict beam pointing stability because the system is extremely susceptible to interference from the mechanical vibration of the loading platform. Severe jitter will cause the laser beam to deviate from the field of view of the target receiving system, resulting in pointing errors that increase the probability of signal interruption and information loss. Precise pointing, acquisition, and tracking (PAT) of incident light to establish a communication link often requires complex optical and hardware systems to account for the relative movement between terminals. For small satellites with strict size, weight and power (SWaP) requirements, the PAT system also increases the system burden. Therefore, it is vital that the miniaturized laser communication precise pointing transceiver system be studied.

Mirror-based PAT systems use reflective mirrors as actuators to achieve beam stabilization, pointing, and tracking. The reflector mirrors deflect the incident laser beam onto the receiving sensor, but unlike traditional gimbal-based PAT mechanisms, mirror-based PAT systems are lightweight, have high response speeds, and provide fine pointing resolution. Feedback control systems are typically used to detect errors in the trajectory or phase of the reflected beam and to fine-tune the deflection angle of the reflector mirror.

Micro-electromechanical Systems (MEMS) is a cutting-edge technology based on micro/nanotechnology [10]. It enables various micro-actuators, sensors, and signal processing circuits to be integrated into a microscale/nanoscale chip. Because of its advantages—small size, ultra-low power consumption, low cost, high integration density, and stable reliability—MEMS is increasingly important for the entire semiconductor ecosystem. It has been widely used in defense, high-speed optical communication, the aerospace and automotive industries, medical equipment, and smart terminal devices [11,12,13].

## 2. MEMS Micromirror Optical Scanning Model

To investigate the relationship between the deflection angle and planar scanning of the 2D MEMS micromirror, a spatial rectangular coordinate system was established (Figure 1). In the initial state, where it did not deflect, the micromirror coincided with the XOY plane, and $\vec{N}$ was the normal unit vector of the XOY plane coincident with the *Z* axis. The incident light $\vec{A}$ formed an angle β with the normal of the MEMS reflection plane on the XOZ plane. The reflected light $\vec{A}\prime$ was reflected by the MEMS reflective plane, entered the X′O′Z′ plane, and was coplanar with it. The distance between *O* and *O*’ was L, the unit normal vector was $\vec{N}$, and the output light was $\vec{A}\prime$. The surface of the MEMS reflective mirror continuously changes as the 2D MEMS mirror vibrates in the horizontal and vertical directions, and this relationship can be derived using the method of matrix optics [12,14].

The vector form of its reflection law can be expressed in matrix optics by Equation (1).
(1)$$\vec{A}\prime =\vec{A}-2(\vec{A}\cdot \vec{N})\cdot \vec{N}$$
where $\vec{A}$ is the incident light unit vector; $\vec{N}$ is the normal direction unit vector; and $\vec{A}\prime$ is the outgoing light unit vector. Transforming the vector form of the reflection law in Equation (1) into a matrix expression, establishes a 3D rectangular coordinate system *x*, *y*, *z* where *i*, *j*, *k* represent the directions of the three-dimensional components in the vector analysis, respectively.
(2)$$\vec{A}=A_{x}\vec{i}+A_{y}\vec{j}+A_{z}\vec{k}.$$
(3)$$\vec{A}\prime =A\prime_{x}\vec{i}+A\prime_{y}\vec{j}+A\prime_{z}\vec{k}.$$
(4)$$\vec{N}=N_{x}\vec{i}+N_{y}\vec{j}+N_{z}\vec{k}.$$

The outgoing light $\vec{A}\prime$, the incident light A and the reflection matrix V can be expressed as Equation (5).
(5)A′=AV.

According to the above description, the expression of the reflection matrix can be obtained as Equation (6).
(6)$$V=\left[\begin{array}{ccc} 1-2N_{x}^{2} & -2N_{y}N_{x} & -2N_{z}N_{x}\\ -2N_{x}N_{y} & 1-2N_{y}^{2} & -2N_{z}N_{y}\\ -2N_{x}N_{z} & -2N_{y}N_{z} & 1-2N_{z}^{2} \end{array}\right].$$

The expression of the reflection matrix V is only related to the component of the normal unit vector $\vec{N}$ of the reflection surface in each direction, and the reflection matrix is determined by the spatial position of the reflection surface. When the MEMS micromirror did not vibrate in the initial state using X′Y′Z′ as the principal coordinate system, the following relationship was obtained:(7)$$\vec{A}=\left[\begin{array}{c} A_{x}\\ A_{y}\\ A_{z} \end{array}\right]=\left[\begin{array}{c} -\sin2\beta \\ 0\\ -\cos2\beta \end{array}\right].$$
(8)$$\vec{N}=\left[\begin{array}{c} N_{x}\\ N_{y}\\ N_{z} \end{array}\right]=\left[\begin{array}{c} -\sin\beta \\ 0\\ -\cos\beta \end{array}\right].$$

Thus, the reflection matrix in the initial state can be obtained as Equation (9).
(9)$$A\prime =AV=\left[\begin{array}{ccc} -\sin2\beta & 0 & -\sin\beta \\ 0 & 1 & 0\\ -\cos2\beta & 0 & -\cos\beta \end{array}\right]\left[\begin{array}{c} -\sin2\beta \\ 0\\ -\cos2\beta \end{array}\right]=\left[\begin{array}{c} 0\\ 0\\ 1 \end{array}\right].$$

Equation (9) confirms that the intersection point of $\vec{A}\prime$ and the X′O′Y′ plane in the initial state is the origin O′, and the outgoing light overlaps with the Z′ axis. When the MEMS mirror undergoes vibrations and deflections in the 2D space, the aforementioned theory will be used to calculate the intersection coordinates of the projection line and the X′O′Y′ plane as well as the deflection angle of the Z′ axis. Assuming that the horizontal deflection angle of the MEMS micromirror is $\alpha_{x}$, the vertical deflection angle is $\alpha_{y}$. This is equivalent to the XOY plane in Figure 1: first rotating $\alpha_{y}$ counterclockwise with the *y*-axis as the axis of rotation and then rotating $\alpha_{x}$ clockwise with the *x*-axis as the axis of rotation. The new normal unit vector corresponding to the reflective surface of the MEMS micromirror can be expressed as Equation (10).
(10)$$\vec{N}=\left[\begin{array}{c} N_{x}\\ N_{y}\\ N_{z} \end{array}\right]=\left[\begin{array}{c} \sin(\beta +\alpha_{x})\cos\alpha_{y}\\ \sin\alpha_{y}\\ \cos(\beta +\alpha_{x})\cos\alpha_{y} \end{array}\right].$$

After calculation and simplification, the output light in the deflected state of the mirror was obtained:(11)$$A\prime =AV=\left[\begin{array}{c} \cos^{2}\alpha_{y}\sin2\alpha_{x}-\sin^{2}\alpha_{y}\sin2\beta \\ \cos(\beta -\alpha_{x})\sin2\alpha_{y}\\ \cos^{2}\alpha_{y}\cos2\alpha_{x}-\sin^{2}\alpha_{y}\sin2\beta \end{array}\right].$$

Regardless of the deviation of the reflecting mirror, the starting point of the output light was always $O^{\prime }(0,0,-L)$. Therefore, the linear equation of the output light was
(12)$$\frac{X^{\prime }}{A_{x}^{\prime }}=\frac{Y^{\prime }}{A_{y}^{\prime }}=\frac{Z^{\prime }+L}{A_{z}^{\prime }}.$$

The coordinates of the intersection point between the incident light and the X′O′Y′ plane was obtained as follows:(13)$$\{\begin{array}{c} X^{\prime }=L\frac{\cos^{2}\alpha_{y}\sin2\alpha_{x}-\sin^{2}\alpha_{y}\sin2\beta }{\cos^{2}\alpha_{y}\cos2\alpha_{x}-\sin^{2}\alpha_{y}\cos2\beta }\\ Y^{\prime }=L\frac{\cos(\beta -\alpha_{x})\sin2\alpha_{y}}{\cos^{2}\alpha_{y}\cos2\alpha_{x}-\sin^{2}\alpha_{y}\cos2\beta } \end{array}.$$

It can be seen from Equation (13) that when $\alpha_{y}=0$, the MEMS micromirror was deflected only in the horizontal direction, and $X=L\tan2\alpha_{x}$ was obtained, indicating that the new reflected line was deflected at an angle of $2\alpha_{x}$ to the original reflected line. This meant that the optical reflection angle in the horizontal direction was twice the mechanical deflection angle of the micromirror. On the other hand, when $\alpha_{x}=0$ the relationship was not satisfied in the vertical direction. Since the condition for this conclusion is that the initial incident light was horizontally incident, it lay in the horizontal plane. When $\alpha_{x}\to 0,\alpha_{y}\to 0$, indicating that the deviation angle of the MEMS mirror was extremely small, we approximated sinα→0,cosα→1, which implied Y/X→cosβ. The equation showed that when the mechanical deflection angle was small, the ratio of the vertical scan length to the horizontal scan length of the reflector was approximately the cosine of the incident light angle; that is, the scanning effect of the MEMS micromirror was related to the incident angle of the initial incident light.

From Equation (13), it is clear that for various horizontal mechanical deflection angles $\alpha_{x}$ and vertical mechanical deflection angles $\alpha_{y}$ of the MEMS micromirrors as well as the different initial light incident angles, the relationship between the horizontal and the vertical deflection angles of the initial output light vector could be expressed as Equation (14):(14)$$\{\begin{array}{c} \varphi_{x}=\tan^{-1}\frac{X}{L}=\frac{\cos^{2}\alpha_{y}\sin2\alpha_{x}-\sin^{2}\alpha_{y}\sin2\beta }{\cos^{2}\alpha_{y}\cos2\alpha_{x}-\sin^{2}\alpha_{y}\cos2\beta }\\ \varphi_{y}=\tan^{-1}\frac{Y}{L}=\frac{\cos(\beta -\alpha_{x})\sin2\alpha_{y}}{\cos^{2}\alpha_{y}\cos2\alpha_{x}-\sin^{2}\alpha_{y}\cos2\beta } \end{array}.$$

## 3. Scanning Range of MEMS Micromirrors

As scanning devices in laser communication, MEMS micromirrors have the significant advantages of being small and giving excellent performance [15]. However, their mechanical deflection angle is relatively limited, and currently available rapid-scanning 2D scanning mirrors typically have mechanical deflection angles of around ±5°, rendering them unsuitable for large-field scanning. In view of this, the expansion angle optical system for the MEMS micromirror is designed to increase the scanning angle. By adjusting the distance between the lens and the micromirror, the optical system expands or reduces the optical scanning range [16].

By designing an optical beam expanding system as shown in Figure 2, the laser beam is emitted and then collimated before being focused onto the reflective surface of a MEMS mirror by a preceding focusing lens. The mechanical deflection angle of the MEMS mirror is relatively small. After passing through an f−θ lens, a light spot corresponding to a certain field of view for the rear lens is obtained. In this case, the field-of-view angle is designed to be 60°.

The maximum image height y on the image plane of the f−θ lens can be expressed as:(15)$$y=f_{1}\cdot \theta_{1}$$
where $f_{1}$ is the focal length of the f−θ lens, and $\theta_{1}$ is 2 times the maximum mechanical deflection angle of the MEMS. Finally, through a positive lens, its field of view $\theta_{2}$ satisfies Equation (16):(16)$$y=f_{2}\cdot \tan\theta_{2}$$
where $f_{2}$ is the effective image square focal length of the angle-expanding lens, and $\theta_{2}$ is the field-of-view angle after beam expansion. Therefore, Equation (17) can be obtained:(17)$$\theta_{2}=\arctan\left(\frac{f_{1}}{f_{2}}\cdot \theta_{1}\right).$$

Obviously, to expand the scanning-angle range of the MEMS micromirror, $f_{2}$ must be smaller than $f_{1}$, which is a key condition of the optical beam expander system. Moreover, when the light emitted by the short-focus positive lens is parallel or close to parallel beams, the laser spot can be made smaller when projected from a longer distance.

## 4. MEMS Micromirror Scanning Capture System

The microsized laser communication terminal scanning system based on MEMS micromirrors is illustrated in Figure 3. The beam emitted by the laser diode is reflected by the MEMS micromirror and then passes through an optical telescope for emission. The optical system magnifies the beam scanning angle. Similarly, the received light is focused onto a photodetector through an optical coupling lens. The detected beam deflection error is fed back to the micromirror controller for real-time correction of the beam deflection angle.

### 4.1. MEMS Micromirror Scanning Mode

Based on the different modes of mechanical vibration, MEMS micromirrors can be classified into quasistatic and resonant types. The former operates at non-resonant frequencies and can be locked at a certain angle or perform vector graphics. The latter operates at resonant frequencies, allowing for larger scanning angles in resonant mode. Depending on the application requirements and fabrication capabilities, both quasistatic and resonant designs can be chosen when designing MEMS 2D scanning micromirrors: dual-axis resonant, dual-axis quasistatic, and resonant with quasistatic. The dual-axis resonant configuration generates Lissajous scanning patterns, but it places higher demands on the resonant frequency compared to the other two at the same spatial resolution. Combining a resonant axis with a quasistatic axis overcomes this drawback and enables grating scanning patterns with larger scanning angles for the resonant axis compared to dual-axis quasistatic scanning. The dual-axis quasistatic configuration is better suited for vector displays and beam control applications.

The three operating modes are primarily designed to address different operational scenarios and requirements. Various applications such as laser projection, LiDAR, and 3D imaging demand fast scanning capabilities without the need for static operation. On the other hand, laser communication requires precise beam stabilization during capture, alignment, and the establishment of the communication link. This requires MEMS micromirrors to operate in a static mode and perform closed-loop tracking based on position feedback. Therefore, in laser communication, MEMS microreflectors work in quasistatic mode for both axes of deflection.

Currently, commonly used methods for laser communication include spiral, raster, and raster–spiral scanning. In this paper, we propose equidistant and equilinear velocity scanning because rectangular raster scanning can cover uncertain areas effectively and is relatively easy to implement. However, it does not prioritize scanning from positions that have higher target–appearance probability. Instead, it treats the entire uncertain area equally, resulting in lower capture efficiency. Moreover, for circular uncertain areas, rectangular-raster and rectangular-spiral scanning have the disadvantage of increasing additional scanning time around the periphery. Spiral scanning is a traditional method that starts from the center point and spirals outward. Its advantage is that it covers uncertain areas quickly, but because the angular velocity remains constant during scanning, the disadvantage is that the linear velocity increases. As the number of spiral cycles increases so does the distance between adjacent scanning points, which leads to missed scans. On the other hand, equidistant and equilinear velocity spiral scanning overcomes this limitation by transforming a constant angular-velocity scan into a spiral scan that has equidistant and equilinear velocity, which improves the beam coverage of uncertain areas. Starting from the center point with the highest target appearance probability, this method scans uncertain area based on the trajectory of the spiral line, resulting in a higher capture probability. Moreover, the scanning pattern conforms to the circular shape of the uncertain area. A schematic diagram of equidistant and equilinear velocity spiral scanning is shown in Figure 4.

### 4.2. MEMS Micromirror Scan Time

In polar and Cartesian coordinate systems, the equidistant and equilinear velocity spiral curve can be represented as follows:(18)$$\left\{ \begin{array}{l} \rho =\frac{I_{\theta }}{2\pi }\theta \\ I_{\theta }=(\frac{\sqrt{2}}{2}-k_{0})\theta_{b} \end{array}\right.$$
(19)$$\left\{ \begin{array}{l} x=\frac{I_{\theta }}{2\pi }\theta \cos(\theta )\\ y=\frac{I_{\theta }}{2\pi }\theta \sin(\theta ) \end{array}\right.$$
where $\theta_{b}$ represents the divergence angle of the laser beam; $I_{\theta }$ represents the spacing between adjacent spiral curves, also known as the scanning step size, which is the distance between two scanning points; $k_{0}$ is the additional overlap factor; ρ represents the polar radius of the scanning spiral curve; and θ represents the polar angle of the scanning spiral curve. Let the time interval between adjacent scanning points be Δt, which represents the dwell time of the beam at a specific point during scanning. In the absence of beacons, the dwell time is only related to the bandwidth $F_{ac}$ of the actuator, denoted as $\Delta t=\frac{1}{F_{ac}}$. Therefore, the spiral scanning speed can be represented as Equation (20):(20)$$v_{0}=\frac{I_{\theta }}{\Delta t}.$$

In conclusion, the expression of the length l of the spiral scanning line in the polar coordinate system can be obtained by the following equations.
(21)$$l=\frac{I_{\theta }}{4\pi }[\theta \sqrt{1+\theta^{2}}+\ln(\theta +\sqrt{1+\theta^{2}})].$$

When the polar angle θ of the scanning spiral curve is greater than π, the length l of the spiral scanning line can be simplified to
(22)$$l\approx \frac{\pi \rho^{2}}{I_{\theta }}=\frac{I_{\theta }}{4\pi }\theta^{2}.$$

Therefore, when the polar angle of the scanning spiral curve is θ, the scanning time t is
(23)$$t=\frac{\Delta t}{4\pi }\theta^{2}.$$

When the scanning polar radius ρ is the half angle $\theta_{u}$ of the capture uncertainty zone, $\rho =\theta_{u}$. After scanning the entire capture uncertain area, the spiral curve reaches its maximum polar angle $\theta_{\max}$ as follows:(24)$$\theta_{\max}=\frac{2\pi }{I_{\theta }}\theta_{u}.$$

The scanning time $t_{u}$ required to scan the entire capture uncertainty region is given as
(25)$$t_{u}=\frac{\pi \theta_{u}^{2}}{I_{\theta }^{2}}\Delta t.$$

For a ±30° uncertain area with a scanning accuracy of 0.1°, a laser beam divergence angle of 1° and a MEMS micromirror scanning bandwidth of 50 Hz, the scanning time can be obtained through simulation, as shown in Figure 5.

From the results, it can be seen that as the half-angle of the uncertain area gradually increases, so does the scanning time. When the area half-angle is 30°, the time is 53.8 s.

## 5. Scan Capture Simulation Experiment

Due to the small divergence angle of the communication light that MEMS micromirrors use for laser communication scanning and capturing, the beam jitter impact is significant. Moreover, the system operates in an open-loop state during scanning, without error correction, which can cause the scanning beam to deviate from its intended trajectory, leading to excessive overlap or missed scanning spots. Ultimately, this results in decreased coverage of the scanning beam over the uncertain area.

Line-of-sight (LOS) jitter is caused mainly by the vibration of the platform. Therefore, this paper proposes mitigating the effect of optical axis jitter by introducing an additional overlap factor during scanning. In general, LOS jitter follows a Rayleigh distribution in the radial direction, and its probability density function can be expressed as
(26)$$f(\alpha )=\frac{\alpha }{\varphi^{2}}\exp(-\frac{\alpha^{2}}{2\varphi^{2}}).$$

During scanning, α is the angular deviation of the scanning-beam jitter φ is the standard deviation of the beam. When the angular deviation α of the jitter exceeds a certain threshold in the azimuth and elevation directions, there is a gap between the scanning spots, resulting in an uncovered scanning area. This phenomenon is known as “missed scanning”. Let’s designate the threshold angle for missed scanning as γ and the probability of missed scanning as $P_{l}$. The expression for $P_{l}$ is
(27)$$P_{l}=\int_{\gamma }^{\infty }f(\alpha )d\alpha =\int_{\gamma }^{\infty }\frac{\alpha }{\varphi^{2}}\exp(-\frac{\alpha^{2}}{2\varphi^{2}})d\alpha =\exp(-\frac{\gamma^{2}}{2\varphi^{2}}).$$

Therefore, the coverage $P_{s}$ of the scanning beam of the communication light on the capture uncertainty region is shown by Equation (28).
(28)$$P_{s}=1-P_{l}=1-\exp(-\frac{\gamma^{2}}{2\varphi^{2}}).$$

From this, the expression of the missed scan-limit angle γ can be deduced as
(29)$$\gamma =\varphi \sqrt{-2\ln(1-P_{s})}.$$

To eliminate the missed scan and meet the scanning beam coverage requirements $P_{s}$ of the uncertain region, an additional overlap factor $k_{0}$ was introduced, as shown in Equation (30):(30)$$k_{0}=\frac{2\gamma }{\theta_{b}}=\frac{2\varphi \sqrt{-2\ln(1-P_{s})}}{\theta_{b}}.$$

To further analyze and validate the impact of factors such as beam divergence angle, overlap factor, size of the uncertain region and possible microvibrations on the performance of laser communication scanning capture, experimental studies were conducted using computer simulations. For this purpose, a GUI experimental interface was designed as shown in Figure 6. This software program allows the flexible adjustment of parameters such as beam divergence angle, overlap factor, detector response time, size of the uncertain region, and standard deviation of microvibrations. By changing the values of these parameters, the scanning trajectory of the spot and its coverage of the uncertain region can be observed in the interface, enabling simulation-based calculations of the time required to complete the scanning of the uncertain region.

First, the simulation program analyzed the scanning coverage of the same uncertain region for different beam divergence angles. The results, shown in Figure 7, indicated that when the system was affected by microvibrations to some extent, certain areas exhibited missed scanning. This observation was consistent with the previous analysis. As the beam divergence angle increased, the time required to scan the entire uncertain region gradually decreased. It was observed that while the microvibrations were kept constant, a larger beam divergence angle led to a smaller area of missed scanning, suggesting that increasing the beam divergence angle had a certain effect on reducing the impact of microvibrations.

Next, while keeping the beam divergence angle constant, the scanning coverage for different sizes of uncertain regions was analyzed. The simulation results, shown in Figure 8, revealed that when the system was affected by microvibrations, changing the size of the uncertain region without altering the beam divergence angle and coverage factor did not reduce missed scans. Moreover, under the same parameter settings, larger uncertain regions required a longer time to complete the scanning. Furthermore, to mitigate missed scanning at the same beam divergence angle, it was necessary to increase the coverage factor, which further increased the scanning duration.

To verify the reduction in missed scanning caused by microvibrations, the overlap factor was gradually increased from 0 to 0.4 while keeping other system parameters constant. The simulation results, shown in Figure 9, indicated that when the system was affected by microvibrations, severe missed scanning occurred if no overlap factor had been introduced. However, when it was introduced and gradually increased to 0.3, there was a noticeable reduction in microvibrations with a standard deviation of 5 µrad. Until the overlap factor reached 0.4, there were no missed scans. However, it is important to note that as the overlap factor increased, scanning time also increased, which should not be ignored. At an overlap factor of 0.4, the time required for the system to scan the entire uncertain region increased by 68% compared to no overlap factor. In practical applications, it is necessary to consider both the scanning time and coverage rate to select optimal parameters.

Finally, to verify the effect of microvibrations on the system visually, standard deviations of 3, 5, 10, and 20 µrad were introduced under the same beam divergence angle, overlap factor, and uncertain region size. The scanning simulation results in Figure 10 revealed that as the effect of microvibrations gradually increased, the occurrences of missed scanning became more severe.

As mentioned earlier, to mitigate the effect of microvibrations, increasing the size of the overlap factor was considered. Building upon the previous set of simulation experiments, the overlap factor was gradually increased up to 0.5 while the microvibration standard deviation remained at 20 µrad. The scanning simulation results at this stage are shown in Figure 11. With an overlap factor of 0.4, there were still some instances of missed scanning, but the beams had already covered the majority of the uncertain region. When the overlap factor reached 0.5, there were no missed scans. However, the scanning time of the system reached 44 s, representing a 95% increase compared to the 0.2 overlap factor.

Based on the above analysis, it is evident that the system inevitably experiences some level of micro-vibration effects. These micro-vibrations introduce missed scanning occurrences during the scanning process, which adversely impact the scanning capture probability. To overcome the detrimental effects of micro-vibrations in equidistant and equilinear velocity spiral scanning, the introduction of an overlap factor is necessary. However, increasing the overlap factor also increases the system scanning time. Therefore, in practical applications, a comprehensive consideration of the overlap factor, balancing capture probability and scanning time, is required. Through simulation analysis, the optimal system parameters can be determined.

## 6. Conclusions

This paper presented a laser beam scanning and capture model based on MEMS micromirrors that focused on fast and wide-range scanning strategies. By introducing a scanning overlap factor, the capture probability under the influence of microvibrations was enhanced. Experimental analysis validated the effectiveness of the scanning capture strategy because platform microvibration was one of the main factors affecting laser communication scanning and capture. When the system experienced certain levels of microvibration, missed scans occurred. As the beam divergence angle gradually increased, the time required to scan the entire uncertain region decreased. Increasing the beam divergence angle while keeping microvibrations constant led to smaller missed scanning areas, indicating the effectiveness of increasing the beam divergence angle to reduce microvibrations. Changing the size of the uncertain region without altering the beam divergence angle or overlap factor did not improve missed scanning caused by microvibrations. Moreover, under the same parameter settings, larger uncertain regions required more time to complete scanning. If mitigating missed scanning occurrences with a constant beam divergence angle were desired, increasing the overlap factor became necessary, but this further increased the scanning time. In practical applications, a comprehensive consideration of scanning time and coverage rate is required to select optimal parameters. The scanning capture model established in this paper provides a theoretical foundation and technical support for practical applications.

## Figures and Tables

**Figure 1 micromachines-14-01317-f001:**
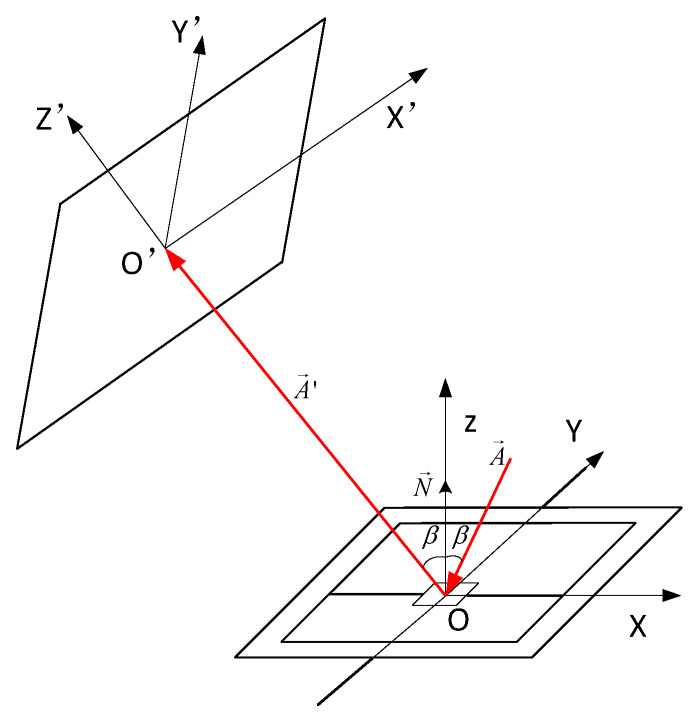
Optical scanning model of 2D MEMS micromirror.

**Figure 2 micromachines-14-01317-f002:**
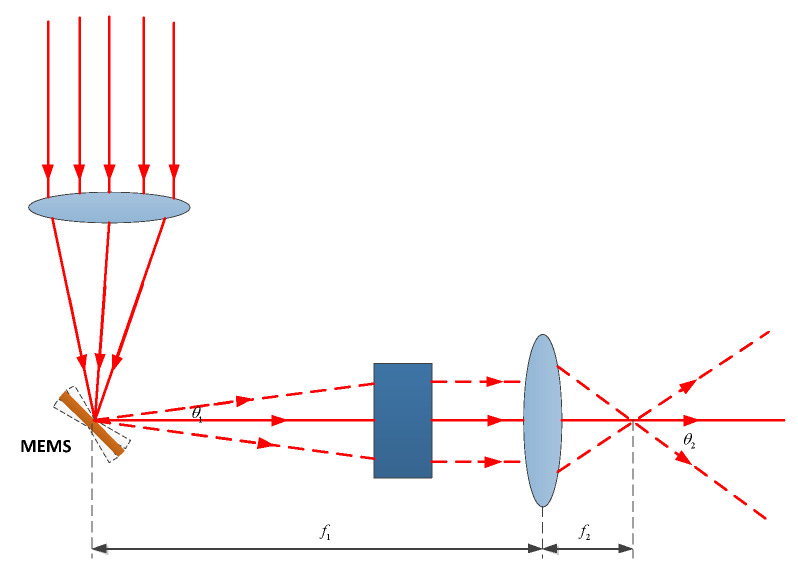
Schematic diagram of the principle of the MEMS micromirror scanning-expansion angle.

**Figure 3 micromachines-14-01317-f003:**
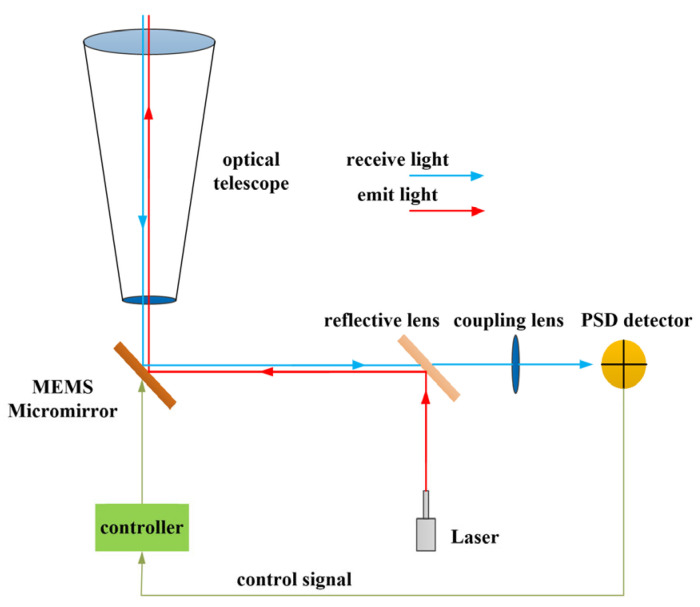
Schematic diagram of laser communication capture and scanning system based on MEMS.

**Figure 4 micromachines-14-01317-f004:**
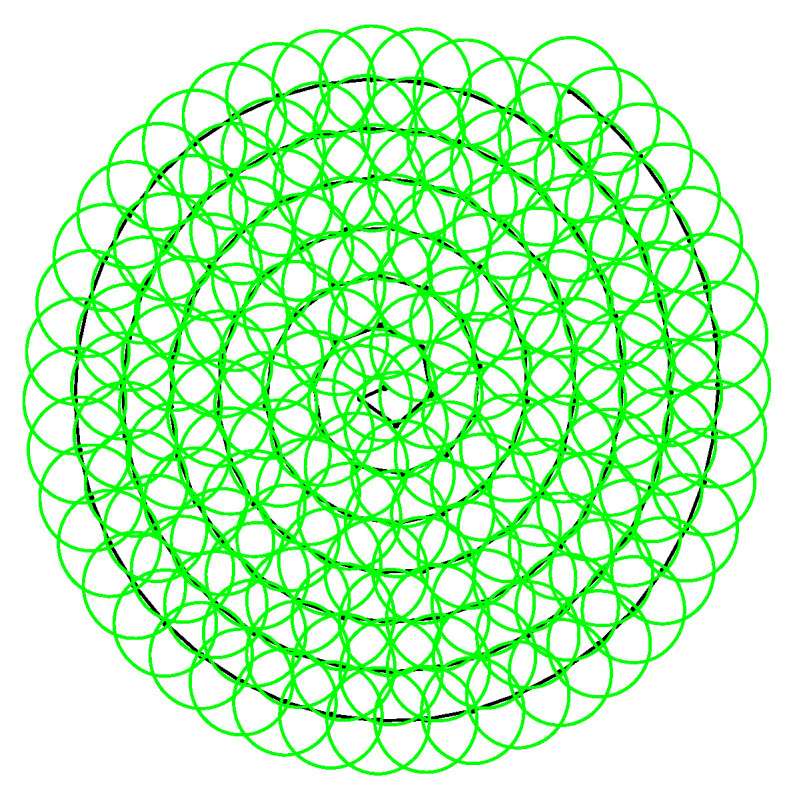
The schematic diagram of equidistant and equilinear velocity spiral scanning.

**Figure 5 micromachines-14-01317-f005:**
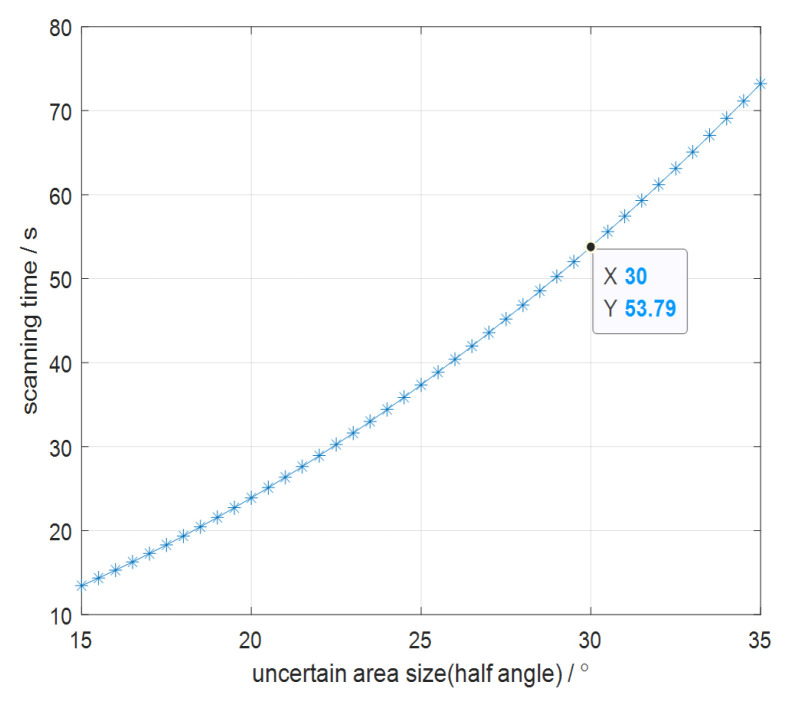
The relationship between the laser beam scanning time and the size of the uncertain region.

**Figure 6 micromachines-14-01317-f006:**
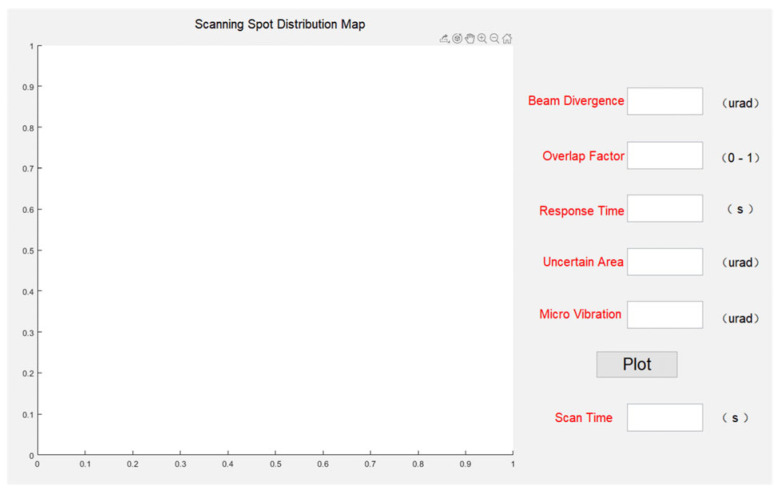
Laser beam scanning capture simulation experiment interface.

**Figure 7 micromachines-14-01317-f007:**
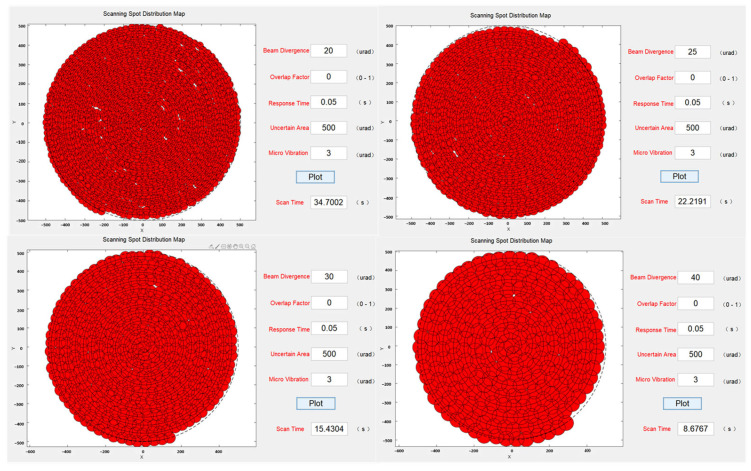
Scanning capture spot distribution at different beam divergence angles.

**Figure 8 micromachines-14-01317-f008:**
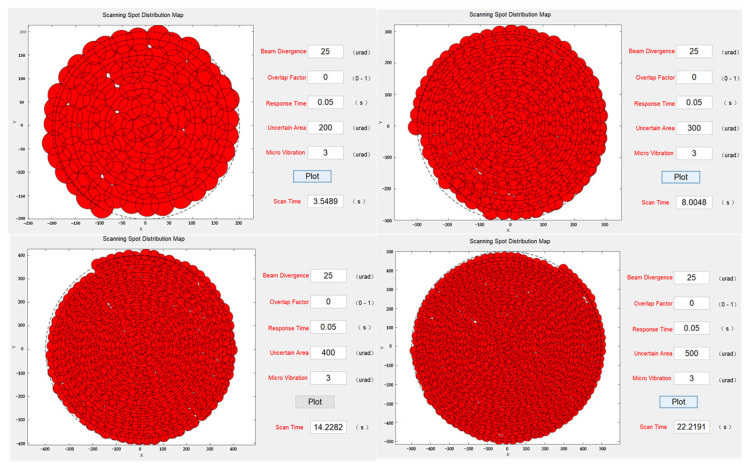
Scanning capture spot distribution at different uncertain regions.

**Figure 9 micromachines-14-01317-f009:**
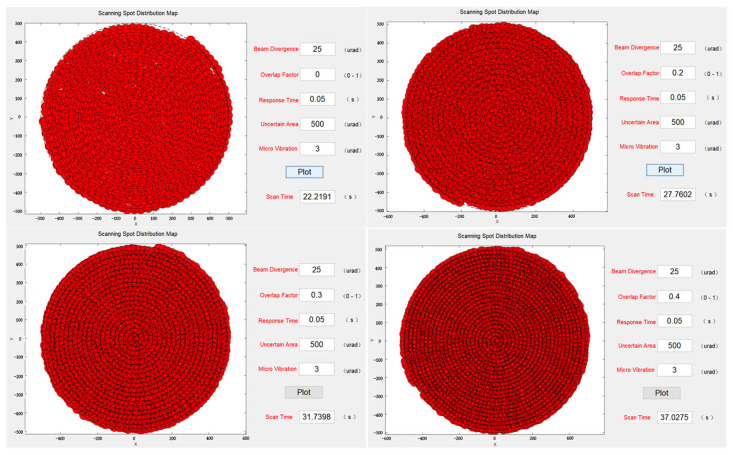
Scanning capture spot distribution at different stacking factors.

**Figure 10 micromachines-14-01317-f010:**
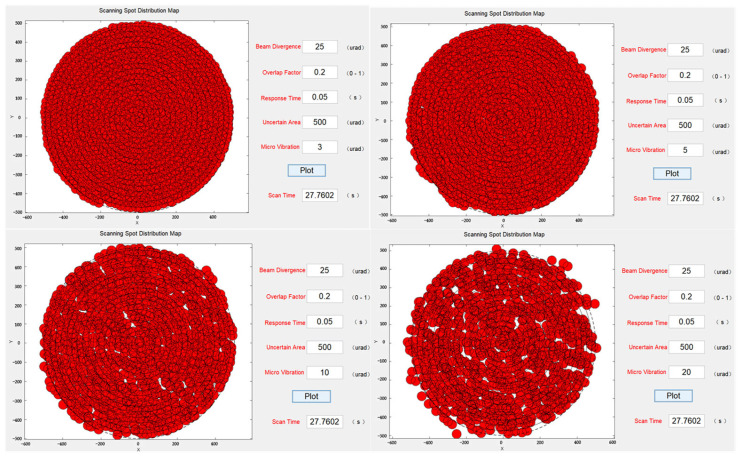
Scanning capture spot distribution under the influence of different microvibrations.

**Figure 11 micromachines-14-01317-f011:**
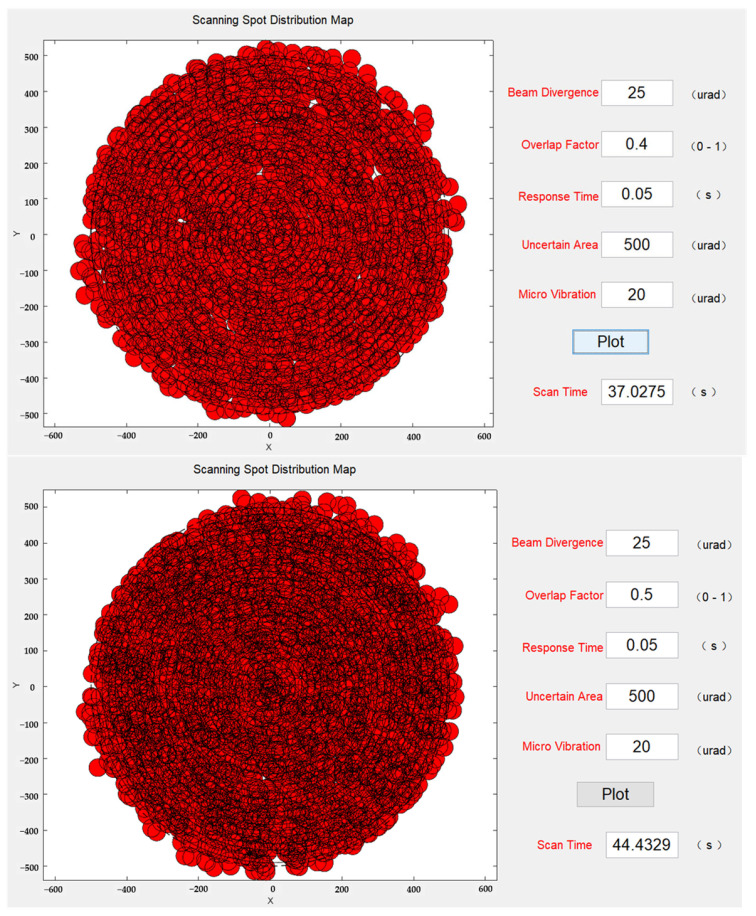
Reduction in missing scans by increasing the overlap factor.

## Data Availability

The data will be made available at a reasonable request to the corresponding author.

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
