# Peer review of "Beam Scanning and Capture of Micro Laser Communication Terminal Based on MEMS Micromirrors"

_micromachines, 2023, doi:10.3390/mi14071317_

Round 1
Reviewer 1 Report
The paper present a theoretical study of the effect of vibration in terms of horizontal and vertical angle components on the performance of laser communication link using spiral scanning approach with MEMS micro mirror. The authors derived the model based on the geometry that overlaps the z axis with the desired direction of light reflection from the MEMs mirror. That allows for easier estimation of the error due to jittering. The paper has good merits in it can be beneficial to the readers subject to some needed clarification and corrections as stated below.
1- In equation 7 it seems that the angles are set relative to OO’ unlike the text description where the coordinates of xyz were used to analyze the vectors. If xyz are used then A=[-sin(b);0;-cos(b)] and N=[0;0;1]. I believe the explanation of the geometry is a bit confusing. A better explanation is needed. For instance the drawing in figure 1 should show the x’y’ plane to be perpendicular to the OO’ vector. The drawing is misleading as it shows the plane to be perpendicular to xy plane.
2- Should not equation 11 yield the same result as 9 when ax and ay vanish? The provided equation is reduced to [0;cos(b),0] instead. Can you explain this?
3- In page 4, line 123, when ax=ay=0 then X=0 and Y=0. Where did you get the ratio Y/X=cos(b)? Did you mean when ax and ay —>0. In other words when ax and ay are very small? Yes in this case with crude approximation the ratio mentioned can be obtained. This section needs proper description in my view in order to make it clear for the reader.
4-In page 4 line 124, when alpha is small sin(a)—> zero but cos(a)—>1 not zero as mentioned in the text.
5- In equation 15,small angle approximation is used as the MEMs scanning range covers 5 degrees. It might be easier for the reader if such statement is mentioned before the equation or simple small angle approximation is mentioned at least. However, personal I do not find a benefit of the approximation at this point as equation 17 can be written in terms of tan(q1) instead of approximating tan(q1) by q1.
6- In equation 18, what is qb (theta_b)? It is not mentioned in the text. Is it a constant?
7-In equation 20, the spiral scanning speed has units rad/sec not m/sec? This is as Iq is in radian. If this is the case, then the derivation in equation 23 is inaccurate. The time is calculated there by dividing l/vo while l is in meter and Iq is in radian.
7- Equation 26, the probability distribution function f is placed there without introduction or the reason of the selection of this particular form.
Reviewer 2 Report
The manuscript proposes a laser beam scanning-capture model based on MEMS micromirror and analyzes a fast and large-scale scanning strategy. Microvibrations introduced missed scanning occurrences during the scanning process in equidistant and equilinear velocity spiral scanning. To overcome this detrimental effect, the introduction of an overlap factor is proposed. However, this increases the system scanning time. The authors propose to optimize the system parameters by balancing capture probability and scanning time based on the scan capture simulation experiment.
The work is interesting and deserves publication in the Micromachines.
I recommend to change "system's" by "system" (lines 251, 258, 358), "equation 1" to Eq. (1) on line 87, "equation 20" by "Eq.(20)" on line 236, "equation 27" by "Eq.(27)" on line 277, "equation 29" by "Eq. (29)" on line 281, and so on.
Round 2
Reviewer 1 Report
The authors have addressed my comments and recommendations.